# Acceptability of a community-embedded intervention for improving adolescent sexual and reproductive health in south-east Nigeria: A qualitative study

Irene Ifeyinwa Eze[1,2]*, Chinyere Okeke[2,3], Chinazom Ekwueme[2,3], Chinyere Ojiugo Mbachu[2,3], Obinna Onwujekwe[2]

**1** Department of Community Medicine, College of Health Sciences Alex Ekwueme Federal University Teaching Hospital Abakaliki, Abakaliki, Nigeria, **2** Health Policy Research Group, University of Nigeria Enugu Campus, Enugu, Nigeria, **3** Department of Community Medicine, College of Health Sciences, University of Nigeria Enugu Campus, Enugu, Nigeria

* jorenebiz@yahoo.com

## Abstract

### Introduction

Adolescents have limited access to quality sexual and reproductive health (SRH) services that are key to healthy sexual lives in many low and middle-income countries such as Nigeria. Hence, context-specific interventions are required to increase adolescents' access to and utilisation of SRH. This paper provides new knowledge on the acceptability of a community-embedded intervention to improve access to SRH information and services for adolescents in Ebonyi state, southeast Nigeria.

### Methods

A community-embedded intervention was implemented for six months in selected communities. Thereafter the intervention was assessed for its acceptability using a total of 30 in-depth interviews and 18 focus group discussions conducted with policymakers, health service providers, school teachers, community gatekeepers, parents and adolescents who were purposively selected as relevant stakeholders on adolescent SRH. The interview transcripts were coded in NVivo 12 using a coding framework structured according to four key constructs of the theoretical framework for acceptability (TFA): affective attitude, intervention coherence, perceived effectiveness, and self-efficacy. The outputs of the coded transcripts were analysed, and the emergent themes from each of the four constructs of the TFA were identified.

### Results

The intervention was acceptable to the stakeholders, from the findings of its positive effects, appropriateness, and positive impact on sexual behaviour. Policymakers were happy to be included in collaborating with multiple stakeholders to co-create multi-faceted interventions

**Funding:** The research project leading to the results presented in the manuscript received funding from the IDRC MENA+WA implementation research project on maternal and child health (IDRC grant number: 108677). The funders had no role in study design, data collection and analysis, the decision to publish, or preparation of the manuscript. The views presented in the manuscript solely belong to the authors and do not represent the funders' views.

**Competing interests:** The authors have declared that no competing interests exist.

relevant to their work (positive affective attitude). The stakeholders understood how the interventions work and perceived them as appropriate at individual and community levels, with adequate and non-complex tools adaptable to different levels of stakeholders (intervention coherence). The intervention promoted mutualistic relations across stakeholders and sectors, including creating multiple platforms to reach the target audience, positive change in sexual behaviour, and cross-learning among policymakers, community gatekeepers, service providers, and adolescents (intervention effectiveness), which empowered them to have the confidence to provide and access SRH information and services (self-efficacy).

## Conclusions

Community-embedded interventions were acceptable as strong mechanisms for improving adolescents' access to SRH in the communities. Policymakers should promote the community-embedded strategy for holistic health promotion of adolescents.

## Introduction

The sexual and reproductive health (SRH) of adolescents is often not given the attention that it deserves in health policies and programmes, especially in low- and middle-income countries (LMICs). This is notwithstanding that adolescence is associated with high sexual and reproductive risks [1], and SRH, which includes access to information, products, and services to manage sexuality with dignity and privacy, is integral to their overall well-being [2, 3].

Globally, nearly 14 million adolescents give birth yearly, and more than 90% of these live births occur in LMICs, with about 21 million pregnancies yearly and half being unwanted [4]. About 55% of unwanted pregnancies lead to abortions, usually unsafe abortions, particularly in countries where abortions are illegal [4]. Furthermore, adolescent mothers have a higher risk of pregnancy-related complications than older women and babies from adolescent women have greater risks of low birth weight and preterm delivery [5]. Complications from pregnancy and childbirth are the second commonest cause of death among older adolescents [6]. In Nigeria, most girls are married and have had sexual intercourse by 19 years [7]. The 2018 demographic and health survey reported that 19% of women aged 15–19 had begun childbearing, rising rapidly from 2% at age 15 to 37% at age 19 [8]. Regardless, the contraceptive prevalence rate was abysmally poor (3%) among this age group [8].

In LMICs, including Nigeria, adolescents face several barriers to accessing SRH services. These are due to poor awareness of SRH, high cost of SRH services, lack of privacy and confidentiality in seeking SRH services, negative attitudes of health providers, and poor appeal of SRH services to adolescents [9]. Addressing challenges adolescents face accessing SRH services through workable interventions is key to achieving global good health and well-being per the Sustainable Development Goal 3 [3, 10]. Although several interventions for improving access of adolescents to SRH information and services have been implemented in LMICs [11, 12], successful SRH interventions are characterized by comprehensiveness, context relevance, the ability to address long-standing myths and misconceptions, and stakeholder acceptance [4, 13].

Acceptability is a major driver of successful implementation and long-term sustainability of interventions for improving access of adolescents to SRH information and services [14]. Acceptability of intervention is *'the perception among implementation stakeholders that a given treatment, service, practice, or innovation is agreeable, palatable, appropriate or satisfactory'*

[15]. Acceptability improves the uptake of an intervention and its effectiveness in the implementation setting. Similarly, those whom the intervention targets are more likely to participate if they consider the interventions appropriate [16].

This paper provides new knowledge on the acceptability of a community-embedded intervention to improve access to SRH information and services for adolescents in Ebonyi state, southeast Nigeria. Exploring the acceptability of adolescent SRH interventions by the stakeholders will provide valuable information that will help create interventions that are more acceptable to the recipients and deliverers of the intervention. Applying a community-embedded approach in this study allows a comprehensive analysis of SRH intervention acceptability outside the commonly studied school-based setting in LMIC [17] by including non-school-going adolescents and community members. This will help in improving uptake and sustained adoption of the intervention. This paper aimed to assess the acceptability of a community-embedded intervention to improve access to SRH information and services for adolescents in southeast Nigeria, using the theoretical framework for acceptability focusing on its four key constructs: affective attitude, intervention coherence, perceived effectiveness, and self-efficacy.

## Methods

### Theoretical framework

Sekhon's Theoretical framework of Acceptability (TFA)—Fig 1, was adopted in this study [15]. It proposes that acceptability is a multi-faceted construct with seven domains—affective attitude, burden, ethicality, intervention coherence, opportunity cost, perceived effectiveness, and self-efficacy, that allow for a robust assessment of overall intervention acceptability by providing an in-depth understanding of the value systems and contextual factors that informs intervention acceptance [15]. Studies have shown the effectiveness and usefulness of the TFA in assessing the acceptability of a comprehensive SRH intervention [17, 18]. Thus, acceptability was assessed along four key constructs of the Theoretical Framework for Acceptability (TFA), namely, affective attitude, self-efficacy, perceived effectiveness, and intervention coherence. Affective attitude refers to *"how the adolescent feels about the intervention"*, self-efficacy is *"the stakeholders' confidence that they can perform the behaviour(s) required to participate in the*

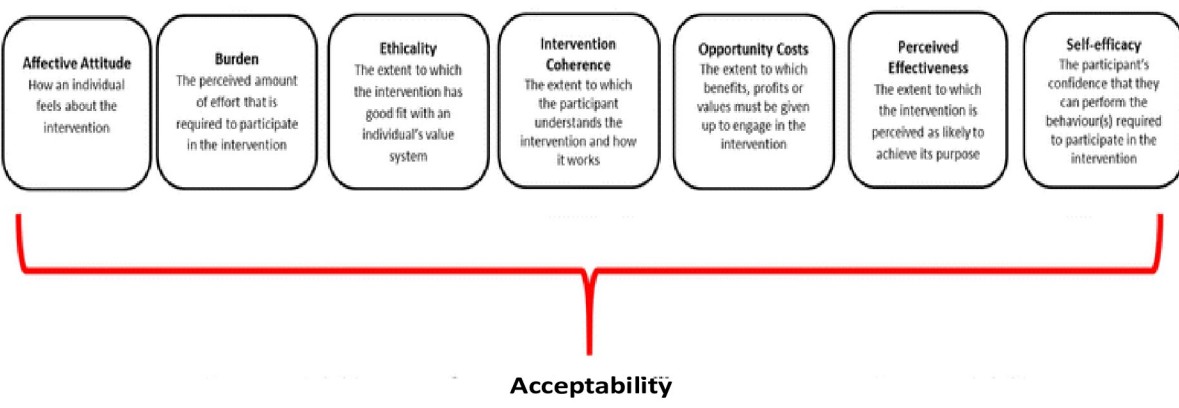

**Fig 1. Theoretical framework of acceptability.**

*intervention"*, intervention coherence is *"the extent to which the participant understands the intervention and how it works" and* perceived effectiveness–"the *extent to which the intervention is perceived as likely to achieve its purpose"* [19].

## Study design and setting

It was a qualitative assessment of a community-embedded adolescent sexual and reproductive health (ASRH) intervention conducted in Ebonyi state, southeast Nigeria. The state is divided into three senatorial zones with thirteen local government areas (LGAs). Most of the populace resides in rural areas, and they are predominantly Christian. Ebonyi State has an estimated population of 2.9 million as of 2016. The State has a fertility rate of 5.4, an adolescent birth rate of 107/1000, and 8.2% of late adolescent girls aged 18–19 years have started childbearing [8]. Health service delivery is structured into a three-tier system, the primary level of care, the secondary and tertiary levels. Overall, the state has two tertiary health facilities, 13 secondary health facilities and six faith-based hospitals, which are engaged in a public-private partnership with the State government and about 631 primary health centres (PHC) (public and private) which render sexual and reproductive health services [20].

## The SRH intervention

The intervention was multi-faceted, involving several components and stakeholders, and targeted in-school and out-of-school adolescents aged 13–18. It was implemented at health facilities, schools, communities, and mass media. It involved several stakeholders that included policymakers, health facility managers, adolescent health focal persons, community health workers (CHWs), patent and proprietary medicine vendors (PPMVs), school principals and teachers/guidance counsellors, peer educators and community leaders, parents, and adolescents.

The components of the intervention were: i) stakeholders engagement through targeted advocacy visits to policymakers and community leaders for support and buy-in using policy briefs, facts sheet, citizen consultations, public panel discussion and policy dialogue, ii) capacity building of the state trainers, health facility managers, CHWs and PPMVs through three-days step-wise training on the provision of adolescent-friendly SRH services, and supportive supervision, iii) training of school teachers/guidance councillors and peer mentor via three-days workshops and establishment of school-based youth clubs for the provision of comprehensive sexuality education, (and reinforced with the distribution of SRH customized sensitization items- notepads, fliers, shirts, caps, wrist bands, pens), iv). Sensitization and awareness creation of community gatekeepers, parents, and adolescents on SRH and rights, and reinforced using similar SRH customized items as per the previous group. Further intervention details can be found in an earlier published manuscript [21].

## Study population and sampling for intervention evaluation

The study population were stakeholders purposively selected due to their relevant in adolescent SRH. They comprise appointed policymakers, career policymakers in health, career policymakers in education, information, youths and sports development, women affairs (boundary partners); healthcare providers, including adolescent focal persons, officers in charge (OIC) of reproductive units in primary health centres (PHCs), community health workers (CHWs), patent and proprietary medicine vendors (PPMVs); principals, teachers, guidance counsellors; traditional leaders, community heads and parents; and adolescents -both in and out of school aged 10-19years and youth advocates across the six intervention clusters. Furthermore, the study population who provided written informed consent were

purposively selected depending on the level of participation in the intervention for inclusion in the study. Participants who were indisposed to communicate due to severe or debilitating medical conditions by the scheduled date for data collection were excluded from participating in the study.

## Data collection methods

This qualitative study was conducted from April 2021 to August 2022 through thirty in-depth interviews (IDI) and eighteen Focus Group discussions (FGDs) with stakeholders involved in adolescent SRH; the sample size at the attainment of saturation. The consolidated criteria for reporting qualitative studies (COREQ) checklist was followed. Semi-structured topic guide adapted from Sekhon's Theoretical framework of Acceptability (TFA) for implementation research that examines seven key constructs: affective attitudes, burden, ethicality, intervention coherence, opportunity costs, perceived effectiveness, and self-efficacy [19] was used (Fig 1).

A total of 30 IDIs were conducted for the following stakeholders: career and the appointed policymakers (N = 16), adolescent focal persons and administrators (N = 8), and school principals/teachers/guidance counsellors (N = 6). Furthermore, eighteen FGDs were conducted, comprising: six FGDs for the healthcare providers- two FGDs each for the three categories of healthcare providers -OICs, CHWs, and PPMVs; two FGDs each for community leaders and parents; and four FGDs each for in-school adolescents and adolescents in community (out-of-school adolescents) The groups were stratified by gender and location—school and community (Table 1).

The interviews/discussions were conducted face-to-face from September 2021 to December 2021 by six research assistants (male and female), well-experienced in qualitative research (social scientist) independent from the implementation team, and communication was in either English or Igbo dialect (as preferred by the participants). Before data collection, the researchers facilitated three days of training for the research assistants on the overview and methodology of the study, principles of research ethics and qualitative data collection, and contents of the research tools (IDI and FGD question guides). The participants were approached through invitation letters and scheduled appointments. The IDIs were conducted individually for the selected stakeholders at their respective offices. The FGD, which comprised 6–8 persons in each group, was conducted in quiet and convenient locations chosen by respondents.

The topic guides explored respondents' perspectives and experiences with the intervention on pre-defined subject areas related to the objectives and acceptability constructs. Prompts were included to ensure that all the relevant aspects of the research questions were exhausted, and notes were taken. Written informed consent was sought from interviewees before the interviews were initiated and for the digital recording. Personal identifiers were redefined to protect the participants' identities, maintain anonymity, and facilitate confidentiality. The FGD lasted 60–90 minutes, and the IDI 45–60 minutes. Observations and note-taking from the SRH intervention meetings were used to gain a deeper understanding of operational issues and the context in which the intervention was conducted.

## Ethical considerations

Ethical approval was obtained from the Research and Ethics Committees of Ebonyi State Ministry of Health (Ref: ERC/SMOH/AI/005/18-21) and the University of Nigeria Teaching Hospital Enugu (Ref: UNTH/CSA/329/OL.5). Approvals were obtained from the school principals. Informed written consent was obtained from participants 18 years and above and parents/guardians of adolescents aged 13 to 17. Furthermore, written assent was obtained

**Table 1. Profile of study participants.**

| Description of participants | Type of interview | | Number of participants | Gender | |
|---|---|---|---|---|---|
| | IDI | FGD | | Male | Female |
| **Policymakers and boundary partners** | | | | | |
| State Ministry of Health | 3 | - | 3 | - | 3 |
| State Ministry of Education | 1 | - | 1 | - | 1 |
| State Ministry of Information | 1 | - | 1 | - | 1 |
| Legislator | 1 | - | 1 | 1 | - |
| State Ministry of youth and sports development | 1 | - | 1 | 1 | - |
| Primary Health Care Development (SPHCDA) | 1 | - | 1 | - | 1 |
| Media (EBBC) | 2 | - | 2 | 1 | 1 |
| SDGs | 1 | | 1 | - | 1 |
| Traditional rulers | 1 | - | 2 | 2 | - |
| Religious leader | 1 | - | 1 | 1 | - |
| NGO | 1 | - | 1 | - | 1 |
| CSO | I | - | 1 | 1 | - |
| **Health service providers/supervisors and teachers** | | | | | |
| LGA ASRH focal officers | 6 | - | 6 | 1 | 5 |
| LGA admin secs | 2 | - | 2 | - | 2 |
| Teachers/Principals/GC | 6 | - | 6 | 2 | 4 |
| PHC workers (OICs) | - | 2 | 7/7 | 7 | 7 |
| PMVs | - | 2 | 6/6 | 4 | 8 |
| CHWs | - | 2 | 7/6 | 6 | 7 |
| **Parents, community leaders and adolescents** | | | | | |
| Male parents | - | 1 | 6 | 6 | - |
| Female parents | - | 1 | 6 | - | 6 |
| Male community leaders | - | 1 | 6 | 6 | - |
| Female community leaders | - | 1 | 6 | - | 6 |
| Male in-school adolescents | - | 2 | 6/6 | 12 | - |
| Female in-school adolescents | - | 2 | 6/6 | - | 12 |
| Male adolescents in-community | - | 2 | 6/4 | 10 | - |
| Female adolescent in-community | | 2 | 8/6 | - | 14 |
| **Total** | **30** | **18** | **143** | **63** | **80** |

from adolescents aged 13 to 17. Consent was obtained by signing the consent form after a thorough explanation, including the benefits and risks of participation, was given, and understanding was established. Participation was voluntary, and respondents were informed that they were at liberty to decline to participate or withdraw from the study with no consequences to them at any time. Confidentiality was assured to participants, and personally identifiable information was not captured. The interviews were held in private and convenient locations.

## Data management and analysis

The audio recording of the interviews (was transcribed verbatim, translated to English, and reviewed with the notes taken during the interview for inductive and deductive thematic analysis. The data analysis followed Braun and Clarke's guide to conducting a thematic content analysis [19]. Initial deductive coding was based on the seven constructs of the TFA, and inductive coding was used to explore new emerging themes that the TFA did not cover. Three transcripts were read through for familiarization and coded manually by three researchers. A

senior social scientist also coded some of the initial transcripts and compared notes with the researchers to ensure coding consistency and comparability and to facilitate collaborative thematic analyses throughout. The transcripts were then imported into NVivo 12 software, Texas, USA. Using the agreed framework, data were coded into pre-defined key themes outlined by the constructs of the TFA, and additional themes/sub-themes were generated.

The generated themes and sub-themes were continually reviewed and refined to capture emerging codes on four key domains of the TFA- affective attitudes, intervention coherence, perceived effectiveness, and self-efficacy. Quotes were captured to highlight the thematic areas and increase our understanding of the context. In-depth discussions between the researchers revealed the more nuanced feelings and perceptions held by participants. Furthermore, the minutes of the intervention team meetings were reviewed and triangulated with the data from the interviews to generate more comprehensive data. The findings were then organized into narratives of the selected constructs of acceptability of the community-embedded SRH intervention.

## Definition of variables

**Independent variables.** These are the socio-demographic characteristics, including gender, age, level of education, designation/position, place of residence etc.

**Outcome variables.** These are the constructs of the theoretical framework of acceptability: affective attitude, intervention coherence, perceived effectiveness, and self-efficacy, defined below [19].

*Affective attitude* was defined as how the adolescent feels about the intervention.

*Intervention coherence* was measured as the extent to which the participant understands the intervention and how it works.

*Perceived effectiveness* was determined by the extent to which the intervention is perceived as likely to achieve its purpose.

*Self-efficacy* was measured as stakeholders' confidence to perform the behaviour(s) required to participate in the intervention.

## Results

A total of 143 persons participated in the study. The participants comprised career and appointed policymakers 16 (11.4%), health services providers 47 (33.3%), school service providers 6 (4.3%), community leaders and parents 24 (17%), and adolescents 48 (34.0%). Majority of the participants, 80 (56.7%), were females, and 61 (42.3%) were males.

Four themes of acceptability were identified, comprising affective attitude with eight sub-themes, intervention coherence with four sub-themes, perceived effectiveness with six sub-themes, and self-efficacy with two sub-themes (Table 2). The qualitative findings are presented in line with four sub-themes.

## 1. Affective attitude

Participants were happy that the intervention allowed them to be included as important partners to collaborate, network, and collectively design robust and multi-faceted intervention strategies relevant to their work and offered learning opportunities.

**i. Happy to collaborate with multiple stakeholders in designing SRH intervention.** A recurrent effect expressed among the policymakers was the feeling of happiness in designing the SRH interventions in collaboration with colleagues from other disciplines and sectors. They were delighted that the advocacy strategy for intersectoral collaboration enabled stakeholders with unique roles to come together to assess adolescents' needs and design

**Table 2. Themes and sub-themes of acceptability.**

| Theme | Sub-theme | Affected study groups |
|---|---|---|
| **Affective attitude** | Happy to collaborate with multiple stakeholders in designing SRH intervention | Policymakers |
| | Grateful to have been included in the design and implementation of the intervention | |
| | Delighted to network with additional stakeholders in the implementation process | |
| | Happy about the robustness of the multi-faceted intervention strategies | |
| | Fascinated with the teamwork and co-creation approach of intervention | |
| | Sad about the limited coverage/reach of SRH intervention | |
| | Delighted with the relevance of training to work | Healthcare providers |
| | Happy about the knowledge learning and sharing opportunities it offered | Adolescents in community |
| **Intervention coherence** | Stakeholders' roles & expectations were clear. | Policy makers, services providers, community members, adolescents |
| | Adequacy of tool—comprehensive and detailed manuals and protocols | |
| | Appropriateness and correctness of tools– <br> • Tools aligned with national guidelines <br> • Incorporation of key sector players and expert—academician and policymakers' perceptive in the development of the tools <br> • Selection of suitable media for intervention implementation and information dissemination | |
| | Information/intervention complexity <br> • Application of non-complex and simple strategies. <br> • stepwise training, <br> • multiple suitable, simple training formats and tools adaptable to different levels of stakeholders | Policy makers, services providers, community members, adolescents |
| **Perceived effectiveness** | Promoted mutualistic relations across stakeholders and sectors | Policy makers, services providers, community members, adolescents |
| | Enlightened on adolescents' SRH and rights and service provision | |
| | Addressed conflict between personal beliefs and societal expectation | |
| | Created multiple platforms for implementers to reach adolescents and eased their work | Service providers, Community influencers |
| | Stimulated implementers' interest in adolescent SRH | Health service providers |
| | Facilitated positive sexual behaviour/lifestyle | Community members, adolescents |
| **Self-efficacy** | It empowered stakeholders to provide SRH education and services to others and have self-worth. | Health services providers, adolescents |
| | Adolescents gained confidence and self-worth in teaching peers | |

comprehensive SRH interventions to improve their health. They expressed that the collaboration was worthwhile as adolescent issues cannot be handled alone as an individual or a sector but need multi-stakeholder involvement. A stakeholder stated:

> *"I am very happy. It* (stakeholder engagement) *was a good aspect of your program because you cannot handle adolescents alone as an individual or a sector. It involves many sectors. For instance, the health ministry can only handle the health aspect very well. They needed the ministry of education to include sex education in school curricula and reach a large population in the schools and the information ministry for awareness creation. In fact, because of Health Policy Research Group, we currently have a weekly one-hour radio slot (every Thursday) to talk about adolescent health*

(IDI, Career Policy Maker-3, Female)

**ii. Grateful to have been included in the design and implementation of the intervention.** The policymakers appreciated that they were recognized and included as relevant partners in improving adolescent sexual and reproductive health. They felt valued that they were

approached from the start of the project and continuously engaged throughout the implementation and evaluation process.

*". . .from the start, I have been part of you; you made me part of the program, so I'm a witness to all that happened. I am very grateful*

(IDI, Boundary Partner-2, Female)

**iii. Delighted to network with additional stakeholders in the implementation process.** Stakeholders were delighted to identify, connect, and network with other stakeholders working on adolescent issues.

*"A few months ago, I got an invitation from you to come for a meeting of collaboration with inter-ministerial persons to advance on the issue of adolescents. I was so happy because I was already in it. . ..and it felt good to know there were more hands in the work*

(IDI, Appointed Policy Maker-1, Female)

*". . . I like the collaboration, and that's why immediately we saw you, we didn't hesitate to welcome you into our fold and partner with you, showing that we are happy that we can see people, agencies, and NGOs work with, knowing the importance of addressing adolescent health issues".*

(IDI, Boundary Partner-2, Female)

**iv. Happy about the robustness of the multi-faceted intervention strategies.** The policymakers expressed happiness that the intervention utilised numerous intervention strategies -advocacy for intersectoral collaboration, stepwise training of healthcare providers, school health program (training of teachers, peer education for peer learning) and community sensitisation, with each strategy having many components. According to them, the multiple components contributed to the program's effectiveness and robustness.

*". . .For instance, the school health program included training teachers/guidance counsellors, and peer mentors, instituting a school health club, supportive supervision . . ..and distribution of information materials the school health club, Yes, I like the multiple approaches. With the inauguration of the school health club, training, and teaching during the process of the inauguration, supervision, and peer learning, I believe the adolescents got the message that we are trying to pass most of them got the message"*

(IDI, Boundary Partner-3, Female)

**v. Fascinated with teamwork and co-creation of intervention.** The policymakers were thrilled about the intervention co-creation strategy, which they described as innovative, fascinating, and interesting. They liked that the process allowed them to brainstorm on relevant thematic areas, which enabled knowledge sharing and developing relevant, context-specific, and useful interventions for improving adolescent SRH.

*". . .the approach was the best . . .. You were able to bring us together, and each team worked in a thematic group which helped to inform the production of the overall materials we used in*

*going into the field. So, to me, . . .it is innovative and interesting. . . because after each day, we normally come out to present what each group has agreed upon that should be enshrined in what we are doing. By so doing, we are sharing knowledge; those in health are learning from information, and all of us are learning from each other. We are celebrating you. You've done a tremendous job".*

(IDI, Boundary Partner-3, Female)

*"I think it* (intervention co-creation) *was a fascinating strategy. . . That's wonderful (smiles) it seems it should not end, but it must still end. . . I hope a new one may be developed where we can mention this one as something that brought up that one. It should spring up another one*

(IDI, Religious leader, Male)

**vi. Sad about the limited coverage of SRH intervention.** Some respondents expressed displeasure at the poor program coverage in the schools. They viewed that the poor coverage may have resulted from fund constraints and recommended collaboration with more partners to scale up the program for all schools.

*I feel sad that such a laudable program did not cover the whole state but was implemented only in six communities and schools. I think we should think of how to get the intervention into all schools.*

(IDI, Boundary Partner-1, Female)

*I think the approach is a welcomed one. . .. Yes, the partnership did it . . .We need more partnerships for scale-up and continuity.*

(IDI, CSO, Male)

**vii. Delighted with the relevance of training to work.** Healthcare workers were delighted that the training was helpful in their work. They expressed happiness that the training was relevant and enhanced their capacity to provide appropriate information and services to adolescents, including referral services on issues outside their scope.

*". . .I am delighted that It (training) impacted me much. None of my customers complained that they missed their period, nor their boyfriends coming to complain that their girls missed their time, and this started after I received my training. I believe they worked with it, and it is helping them.*

(FGD, PMV-R02 EZZ community, Male)

*"I have been using it to educate my younger ones about their rights, what to do and what not to do. This is the first training I have been impressed that I learnt a lot. It's been helping me in my shop most of the time; our adolescents come and interact with you and tell you their problem, you might not know how to handle it, but since then (the training), I have handled it very well. You see them very happy because they can bring another person to you"*

(FGD, PMV-R01, EZZ community, Female).

*They* (adolescents) *are so happy, and I am impressed, to be honest with you, how they respected me before the program came into existence, and now they added more respect to it. Some will tell me, mummy, everything you were teaching us and what we hear here is related. I told them I didn't just stand up from my house one day and start saying what I didn't know. I am also happy because you people do make out time to come on your own to see that what we are telling them is not our order; we didn't read them up from any book; it was you people who came and trained us and then in return we are giving it back to them, to be honest, this thing you people did will surely bear many fruits. So, it's a good one*

(FGD, CHW-R02 OHA community, female)

**viii. Happy about the knowledge-learning and sharing opportunities it offered.**   Adolescents were happy because the health information received provided opportunities to learn about SRH. Some of the adolescents were happy that the intervention empowered them to provide education to their peers and be agents of positive change.

*I like it as it is an opportunity to learn many things. It also helps me to talk to my friends and others so that we know how best to take decisions when anything* arises

(FGD, Adolescent in community-R01, NWF, Female)

*"I feel happy as am a member of this group because of the many things we were taught that we can teach our sisters and* parent"

(FGD, adolescent in community-R02, NWF, Female)

## 2. Intervention coherence

**i. Stakeholders' roles and expectations were clear.**   Stakeholders described their understanding of the intervention and implementing strategies.

*"Yes, it is very relevant and valuable because it helps them and helps us. For example, when there is sexual violence in the community, like "disvirginity" they normally call us. We follow the case up to the hospital where she will be laboratory tested and know if the girl is not infected with HIV, you have to rule out other sexually transmitted diseases, it is our duty as a community health service provider working in that area to follow the case up, till the end, and now we know what to do.*

(FGD, OIC-R06, NWF community, Female).

*". . .Action Research is about discovering the problem and how to address it. It should be something that can be repeated somewhere, and it gives a good result that will lead to a solution. . .. You, people, have done well, and if am here as an assessor, I will give you people an "A" all around*

(IDI, CSO, Male)

**ii. Adequacy of tool—Comprehensive and detailed manuals and protocols.** The information was reported to be comprehensive and detailed by stakeholders.

*I was part of the development. So, I believe the manual was detailed in relation to the target group. . . It was carved for trainers and learners. both were good.*

(IDI, Career policymaker-R01, Female)

**iii. Appropriateness and correctness.** The stakeholders reported that the tools were appropriate and credible, and this they attributed to their adaptation from the national guidelines **and i**ncorporation of multiple key sector players—academicians and policymakers in the development of the intervention tools.

*". . ... It (the tool) is good; it is expected, though, as the training manuals we used are captured from the national training manual.*

(KII Career policy maker-1, female)

*I know you people who developed the manual (laughing) are coming from the university— academicians and all the rest we, the policymakers, and other organisations who collaborated with you, all made input. . . It is good*

(IDI, Career policy maker-2, Female)

Other stakeholders attributed the appropriateness to the use of the suitable media in implementing the interventions and for information dissemination.

*"One thing is that most of our people listen to the radio, and a sizeable number of the citizens have radios; they tune in to the radio. So those whom we could not contact during the advocacy get the information when they tune in and listen to the radio and get the information. So, it was appropriate.*

(IDI, Community influencer, Male)

**iv. Information/intervention complexity.** Stakeholders agreed that the tools were not complex but rather simple and comprehensible. They attributed the non-complexity to the methodology approaches whereby the tools were adapted to the different levels of stakeholders.

*Stepwise training approach.* The policymakers and health workers viewed that the stepwise training approach from the high-level to low-level health workers was good and contributed to the non-complexity of the intervention implementation

*". . .the training method was very good. You remember we were trained as state trainers, and we trained the OICs, who then trained the community health workers and patent medicine vendors, and we all also supervised in that manner. It made the work less complex*

(IDI, Career policy maker-1, Female)

*Multiple suitable and adaptable training formats.* Other stakeholders attributed the non-complexity to using multiple suitable training formats and materials/tools. Furthermore, the

stakeholders stated that using multiple and simple teaching tools, flip charts, lectures, posters, and availability of working tools—training manuals, teaching aids, registers etc., enabled intervention success and ensured that each approach was well suited to the different trainees.

*"it* (training manual) *is handy and simple to carry and discuss, unlike the national training manual, which is very voluminous and not easy to carry and use at the level of community health workers. The one Health Policy Research Group developed is easy to carry, simple to discuss and understandable* to community health workers.

(KII, Career policy maker-1, Female)

*". . .Yes, the manuals and the protocols were of good help to use, even to the students. They got the manuals to read and know what the school health club is about. It made some people know why some of these things are necessary Also, the Manual and the flyers are very useful to me as an individual and as a desk officer because my children's school was not selected, so whenever they are going to have a school health club in their school, they have gotten the information of what school health club is all about.*

(KII, Boundary partner—1, Female)

*"..that teaching aid and flipchart were useful, you know we used to walk around the community, and we go with the teaching aid so that when you teach them, you use it to teach them, and you will also be showing them the pictures in the teaching aid and posters. The manual is really useful in the sense that when you finish reading it, you will now know what to teach other people you met on the way concerning adolescent sexual reproductive health needs.*

(FGD, CHW-RO3 AMG, Female)

## 3. Perceived effectiveness

**i. Promoted mutualistic relations across stakeholders and sectors.** Stakeholders viewed that the intervention enabled beneficial collaboration between relevant stakeholders. Before intervention implementation, there was no functional and organised forum for stakeholders to come together and discuss adolescents' sexual health. Previously community leaders were not opportune to collaborate and brainstorm with experts on sexuality issues. Most parents lacked the skill to communicate information on sexual health and rights to young persons. Bringing together various categories of stakeholders' policy makers, research experts, religious and traditional leaders, parents, and young people created a bond of trust among the stakeholders.

*"On that collaboration particularly, I will say it's nice because our communities don't have a forum where they talk about the sexual and reproductive health rights of adolescents. If you look at parents, they barely talk about it, so even if they do, it is still somehow as a child cannot feel comfortable meeting the parents to discuss it. So bringing in different stakeholders, the religious leaders, teachers, families and other people they believe they can fall back to, listen to, and talk to make everybody feel good . . . it was a partnership that did wonders*

(IDI, CSO, Male)

*"it is very appropriate and relevant. . .I will make sure they (PPMVs) become part of the primary healthcare system so that we work together.*

(IDI, Appointed policymaker, Female)

**ii. Enlightened on SRH services and rights.** There was unanimous agreement among the different state stakeholders that the intervention helped enlighten them about sexual and reproductive health and rights and enhanced access to SRH information and services. These facilitated improvement on adolescents' assertiveness on sexual matters, negotiate and practice safe sex, and boldly seek for SRH information and services.

*"Yes, it* (public panel discussions *was relevant because it brought about ideas and it brought about result-oriented output*

(IDI, Appointed policy maker-1, Female)

*"So, the intervention of this project in our different schools is an eye-opener for the students. From that program, they have been able to know their rights; know when to say no and mean it and be bold to say no, so if anyone is molesting them, they will be bold and say no! no!!!, I don't want it I don't want anything to stop their education, so I thank God for the program; the way you people made it compulsory for the school health club to be in every school participating*

(IDI, boundary partner-3, Female)

*"We cannot dispute that some of the things they were educated on -the challenges and the ways forward are of paramount importance The program (public panel discussion) opened the eyes of the students, and from their questions, it was obvious that it was the need of the time. A lot of them asked questions, and the types of responses they got from the experts-important formation shared, brought the need that this program should be extended to all areas, not just the little group, . . .the radio program) was good. As we talked, most people who listened to the radio contributed and made a great impact. I believed that with the group work of all stakeholders and other projects, we would save these young ones.*

(IDI, community influencer-3, Male)

In addition, parents and adolescents perceived that the intervention was beneficial as it allowed the young persons as well as the parents to learn the correct things from the right source.

*". . .it is of great benefit to the children because they have knowledge of all parts of their bodies and what they represent and know how to take care of sensitive parts of their bodies to avoid problems. . .. and it helped in exposing the adolescents to the truth; our children learn many good things from the campaign; they have known what they are supposed to do and what they are not supposed to do they indeed know about some of these things, but they might be getting the information from the wrong sources, with this campaign they have known the truth, if they happen to get information about reproductive health from any other source, they will weigh it then decide on the right one to choose It is also appropriate for us because it has exposed us to many things that we did not know before"*

(FGD, Parent-R02 NWF community female)

*"It is good; They taught us that it is wrong for boys and girls to have unsafe sexual intercourse.*

(FGD, adolescent in community, AGB Female)

**iii. Addressed conflict between personal beliefs and societal expectation.** Stakeholders reported that the intervention enabled them to freely discuss sexuality, and young people now voluntarily and boldly seek sexual and reproductive health information and services.

*"When you talk about the values there, I had a strict upbringing and my life perception. I find it difficult to tell my daughter about family planning and abstinence and ask her are you still a virgin. It was difficult telling my daughter that if she must do this, she should use a condom. I couldn't, It was as if I was helping them to be promiscuous, both religious value, social value and domestic value-wise. . . but with your intervention,. . . and how we handled it in this program, you have made it possible for me to talk about it.*

(IDI Appointed policymaker-1, Female)

*"..so it has affected their way of life and the norms. When you start talking about such, especially adolescent issues, they feel why you go to that area. They believe that area could be treated with silence, but I told them "No", anything you don't teach people, they will not understand what it is. . .. .., now they are getting used to it. I have told you that's what their parents used to say now "Eze told us last time that you don't associate with these boys; atuo gi Ime, (if you get pregnant) and if he punishes you, people don't call us" they are changing now*

(IDI, Community influencer-2, Male).

**iv. Created multiple platforms for implementers to reach adolescents and eased their work.** Community stakeholders agreed that the intervention was helpful as it created platforms which enabled them to reach large and different categories of the target audience and eased their work. A religious leader stated:

*"it's a wonderful program. It was the most wonderful avenue for us to reach out to the ones we have been looking for over the years, so for me, the organisers provided us with platforms to share our desires to get our youth corrected"*

(IDI, community influencer-3, Male).

A traditional ruler stated:

*"Yes, I know you were with us, the traditional rulers, sometimes in connection with the program. I know we gave you blessings. We bought this idea wholesale and asked you to carry on, it is indeed a good thing it is a way of cutting down our job shorter. So, we gave our blessings to it".*

(IDI, Community influencer-1, Male)

**v. Stimulated implementers' interest in adolescent SRH.** Stakeholders perceived that implementers like the patent medicine vendors were interested and enthusiastic about the intervention and eager to participate.

*"It was very, very interesting; I could remember when we took our supportive supervisions to PPMV... some of them, because we were still somewhere, were already calling, "ma, we are still waiting, oh! I hope you will not disappoint us. We want to show you what we are doing, whether it is how we were asked to do it. This is to show you that they were interested, they were waiting, and they were eager to learn"*

(IDI, Carrier policy maker-1, Female)

**vi. Facilitated positive sexual behaviour/lifestyle.** The intervention was perceived as having facilitated a positive change in SRH behaviour at the individual and community levels. According to the adolescents, the sexuality education, campaigns, and sensitizations enabled them to understand and avoid/reduce risky sexual behaviours such as having multiple partners, transactional sex, alcohol/drug abuse, sexual violence risk factors such as moving around in isolated places and keeping late nights, in addition, it enhanced their self-will to access to SRH information and services including contraceptives like condoms which reduced the rate of teenage pregnancy and sexually transmitted diseases and motivated positive sexual behaviour such as abstinence, Adolescents expressed:

*"I am very happy about the campaign because some bad things have stopped. and there are great and positive improvements in our lives; young girls and boys in this community stopped moving about at 8:00 at night, and some young boys used to pregnant young girls in this community, but it is no longer happening. if not for the campaign, many young girls would have been pregnant by now; the campaign is a good one*

(FGD, adolescent in community, AGB, Male)

*"It* (the campaign*) made me happy because anytime I want to do those things that we used to do before, I will remember it and withdraw* [abstaining from casual sex]. *Many young girls were getting pregnant before now, but it has reduced very well; we learnt many things from the campaign... we are behaving very well now, and things have changed"*

(FGD adolescent in community-R02, NWF, Female)

Other significant adults in the community—perceived the intervention was beneficial as it facilitated a positive sexual lifestyle and reduced the negative consequences of risky sexual behaviour among adolescents, and enhanced parent-child communication including support to access needed SRH services.

*"...The benefit is quite enormous; it helped in many ways to eradicate sexually transmitted diseases and unwanted pregnancies and most of the bad sexual activities and other things–*

(FGD Parent R01, NWF, Male)

*"The campaign is beneficial to all of us in this community, it reduced the rate at which girls are pregnant. It also made the parents know how to discuss with their children. Before now, we send out any of our daughters who get unwanted pregnancies, but now we don't send them out'*

(FGD parent-RO3, NWF Female)

*"The adolescents have been putting what they learnt [sexuality education] into practice because of how and manners they now behave amongst their peers and elderly ones in the community. The truth is that the awareness has been bearing good fruits in the community, and we know that the benefits will continue to be evident in the community"*

(FGD, Community leader-R01, AGB, Male)

*"As a grass root person, I assure you it is relevant now, from what I have observed, most of the girl children voluntarily go to get counselling and SRH services from some of our health facilities, unlike what it used to be before, they used to shy away, but now they can boldly come to the health facility to ask one or two questions and get services and commodities like condom which they need. So, it has been eye-opening for them to be on the right side to say no and mean it".*

(IDI, CSO, Male)

## 4. Self-efficacy

Stakeholders revealed that the intervention affected their agency. It empowered them to provide SRH education and services to others and have self-worth. According to the health workers, because of the training, they are enthusiastic and confident that they could provide SRH information and services.

*"Not only that we concentrate only on the schools benefiting from the intervention, but I will be going to different schools in Ebonyi State to make sure that this information is being spread in every part of this our schools...."HPRG has given us the knife"* (enhanced our capacity), *so as far as our students are concerned, we will continue to do anything to secure our students,*

(IDI. Boundary partner -1, Female)

*"Yes, we are very happy because, with the skills we acquired, we can give the services needed from us, the ones we can afford to give. The referral form: it usually helps us in referring if we have any complicated cases to FETHA or mile 4 hospitals. We now have more confidence in doing the work, as the little effort we are putting into the service is yielding many fruits"*

(FGD, OIC-R05, Female)

*"I am interested in doing it, as a Christian and also as a father; I know what is going on in society today, and I know that the young ones need these services; I have already developed a passion for doing the work, and I will be teaching those that are inexperience in the areas of sexuality education, so I still have the zeal to continue"*

(FGD, CHW-R02, Male)

Adolescents reported that the interventions helped them gain confidence in teaching peers and develop self-worth.

*"I feel happy because I am a member of this group* (school health club) *because we were taught many things that we could teach our sisters and parent. I feel free to discuss with my*

*younger ones; they all put my teaching into practice. Before the campaign, I thought that such teaching was only for mature people, but I have understood that it is good for everybody. We discuss it wherever we find ourselves, I do not feel shy to teach others because I know that it is very important for all of us"*

(FGD, adolescent in community, NWF, Female)

*"I am happy that I can teach my friends about SRH. The person I talked to returned to tell me that he learnt good things from my conversations with him, which has changed many things in his life. This encouraged me even to continue talking to people"*

(FGD, adolescent in community, AGB male)

The In-school adolescents expressed happiness that the intervention enhanced their confidence. According to them, the school health club was an eye-opener as it helped them express themselves freely, unlike the usual school Guidance and Counselling. The adolescents were happy that the intervention made them feel bolder and more relevant, increasing their capacity for better decision-making.

*"I am so glad and excited about the school health club, and I see it as a very interesting one that helps us young girls to express ourselves about what we come across in our homes and school, which we do not get in our G and C guidance. . . it has opened my eye"*

(FGD, adolescent in-school -R03, Female)

*"I'm very happy about it because it has made me bold. . .it makes me feel relevant. It has made me valued because, in the past, I hardly stood firm and made decisions, but now I can stand firm and decide on my own"*

(FGD, Adolescent in-School -R01, female)

## Discussion

Overall, the SRH intervention was acceptable among policymakers, healthcare service providers, community members, and adolescents, especially due to the inclusion of relevant partners to collectively design appropriate and comprehensive multi-faceted strategies adaptable to different levels of stakeholders. Community members accepted the intervention as it enabled mutualistic relations across stakeholders and sectors, which helped address the conflicts between personal beliefs and societal norms and expectations and the sex educational needs in the communities. Furthermore, adolescents welcomed the intervention as it enhanced access to SRH services from multiple and preferred platforms and empowered them to have self-worth and confidence to provide sexual education. These findings support similar qualitative work that assessed the effectiveness and acceptability of a comprehensive menstrual health intervention program in Uganda and Zimbabwe [17, 18].

For most adolescents, the intervention acceptability was heavily informed by the intervention's effectiveness in enhancing access to quality and dignified information, services, and product. This is important and integral to the well-being of adolescents as it prevents ineffective alternatives and harmful practices and lifestyles [22]. The intervention was accepted in the community because it helped address the conflicts between personal beliefs and societal norms and expectations. This is in contrast to a study in Zimbabwe [17], where despite implementing

a community-based SRH intervention for healthcare providers and product sensitization, there were still challenges to overcoming the sociocultural barriers to SRH product acceptability. Concerning the healthcare service providers, the SRH intervention was welcomed as it was perceived as "a hook" to attract adolescents, both in and out of school, and expose them to quality SRH information and services using varied formats. Our study's acceptance of the intervention corroborates earlier studies on the acceptability of SRH products in Zimbabwe [17] and Malawi [22], where similar acceptance was reported. Given the acceptability of the strategies, SRH interventions should be designed to be context-specific, adaptable to the users' needs and preferences, and SRH education prioritised for informed sexual choice.

The appropriateness, non-complexity, and adaptability of the tools to different levels of stakeholders were positive factors that enhanced the acceptability of the intervention. Our co-creation approach created opportunities for key stakeholders to collectively formulate, design and implement workable strategies adaptable to their context, contrary to the view that many novel intervention models fail to address unintended consequences such as how the intervention implementers feel about delivering the intervention [23, 24]. The providers predominantly viewed the co-creation-related work as "innovative" and "enlightening" rather than burdensome, as it provided the opportunity for knowledge sharing, eased their work, and created multiple platforms to reach adolescents. The perception of the co-creation approach as non-burdensome is noteworthy, as evidence has demonstrated that perceptions of burden inform providers' acceptability of interventions [25]. This is important as successful integration models must consider and address staff perceptions about their roles and tasks, especially in settings where healthcare is often delivered through vertical programmes.

Our findings offer evidence of the acceptability of a collaborative learning approach to improve knowledge and skills among health service providers, strengthen the friendliness of SRH services provided to adolescents, and improve communication and trust between health service providers, adolescents, and community members. Our finding aligned with the evidence on the 'client-oriented, provider-efficient' (COPE) process for quality improvement where collaborative learning approaches were used in Congo, and it improved health service provider performance in many areas, including respectful care, information provision, personal communication, privacy, and clinical skills [26].

Our study points out the importance of enabling service providers to learn from and support each other through discussion and reflection. It highlights the need for a multi-sectoral approach to address the wider social determinants of adolescents' health. This view is also buttressed by the substantial male involvement (including traditional leaders) noted in this study which may have contributed to the intervention acceptability in a male-dominated society, contrary to existing evidence on poor male engagement with SRH services [17, 25]. Thus, the need to leverage the stimulated interest in SRH among men to enhance health-seeking behaviours.

Despite the clear acceptability, our findings show that external factors such as funds limited the intervention coverage, and the collaborative learning strategies may face sustainability challenges, including clarity about continuous funding, a view also shared in an earlier study [26]. However, the benefits of the informal sharing and learning processes in this context, notwithstanding the novelty, build on the evidence that collaborative learning approaches may address the challenges in ensuring quality SRH services for adolescents [25, 26]. Indeed, the experiences with collaborative learning highlight the need for a comprehensive approach to performance improvement that goes beyond training to improve health service provider knowledge, skills, attitudes, and motivation. Collaboration between health and non-health sectors may help ensure the longevity of the approach. Collaborative learning should be

incorporated into existing health service plans and budgets. Efforts should be made to ensure programme ownership and scale-up of the approach to other adolescent health services.

The strength of this study is the assessment of acceptability from the perspectives of the intervention clients and providers on how the intervention was implemented, perceived, and experienced. Additionally, by applying a theoretical framework of acceptability which addressed the gap in utilizing an effective model for integrated SRH service evaluation [24, 27], our study provided an in-depth understanding of how individual and community value systems and contextual factors within communities informed acceptability. Furthermore, the community-based setting allows for a much-needed analysis of SRH intervention acceptability outside of a school-based setting and among non-school-going adolescents in an LMIC. The study utilized robust methodology and broad involvement of stakeholders; thus, the findings can be applied to a similar context in the LMIC.

## Strength and limitations

Noteworthy is that the collaboration between health and non-health sectors created multiple platforms to better reach of adolescents with SRH services and facilitated knowledge learning and sharing, mutualistic relations and networking across stakeholders and sectors which would ensure longevity of the approach and ownership of the programme. Despite the strengths, a major limitation of the study is that the data may be subject to social-desirability bias, where respondents, particularly health provider and adolescents, may have felt obliged to report positively on the SRH intervention. However, the research assistants were well-trained to disassociate themselves from the implementation team and probe for all positive and negative opinions.

## Conclusions

Finally, the study revealed that community-embedded interventions are strong mechanisms for improving access to SRH by adolescents in communities and should be promoted for holistic health promotion of adolescents. There were high levels of acceptability of the intervention among the policymakers, service providers, community members and adolescents. These were expressed through their happiness with the collaborative and multi-faceted implementation approach, adequacy, appropriateness, and non-complexity of the implementation tools, capacity and knowledge-sharing opportunity and the positive change in the value system and agency. Given the above, it is recommended that the stakeholders consolidate the gains and acceptance of the collaborative intervention strategies and scale up the intervention. Furthermore, to mitigate the perceived fund constraints, collaboration with more partners is recommended for the continuity of the programmes.

## Supporting information

**S1 File.**
(DOCX)

**S1 Checklist. STROBE statement—Checklist of items that should be included in reports of observational studies.**
(DOCX)

## Acknowledgments

We thank the management and leadership of the institutions–Ministry of Health, State Primary Health Development Agency, Schools, and Communities in Ebonyi State for granting us permission and all the study respondents for their active participation and willingness to participate in the survey.

## Author Contributions

**Conceptualization:** Irene Ifeyinwa Eze, Chinyere Ojiugo Mbachu, Obinna Onwujekwe.

**Data curation:** Irene Ifeyinwa Eze, Chinyere Okeke, Chinazom Ekwueme, Chinyere Ojiugo Mbachu, Obinna Onwujekwe.

**Formal analysis:** Irene Ifeyinwa Eze, Chinazom Ekwueme.

**Funding acquisition:** Chinyere Okeke, Chinyere Ojiugo Mbachu, Obinna Onwujekwe.

**Methodology:** Irene Ifeyinwa Eze, Chinyere Okeke, Chinazom Ekwueme, Obinna Onwujekwe.

**Project administration:** Chinyere Ojiugo Mbachu, Obinna Onwujekwe.

**Supervision:** Irene Ifeyinwa Eze, Chinyere Okeke, Chinyere Ojiugo Mbachu, Obinna Onwujekwe.

**Writing – original draft:** Irene Ifeyinwa Eze.

**Writing – review & editing:** Chinyere Okeke, Chinazom Ekwueme, Chinyere Ojiugo Mbachu, Obinna Onwujekwe.

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
