## [Decision Letter · Decision Letter 0]

9 Oct 2023

PONE-D-23-09502Acceptability of a community-embedded intervention for improving adolescent sexual and reproductive health in south-east Nigeria: a qualitative studyPLOS ONE

Dear Dr. Eze,

Thank you for submitting your manuscript to PLOS ONE. After careful consideration, we feel that it has merit but does not fully meet PLOS ONE’s publication criteria as it currently stands. Therefore, we invite you to submit a revised version of the manuscript that addresses the points raised during the review process.

We look forward to receiving your revised manuscript.

Kind regards,

Alfredo Luis Fort, M.D., M.Sc., Ph.D.

Academic Editor

PLOS ONE

Journal Requirements:

Additional Editor Comments (if provided):

Thank you for your submission. You will see from the reviewers, as well as from my attachment, that there are several sections that need improving or clarifying statements or concepts. This is very important for the reader to make full sense of what they are reading.

However, there are other aspects which need revision. For example, there is no insight into any of the outcomes of the project, even if they are for a short term, and are of things like more acceptability, or people are more aware, or there is better/more communication, etc. Otherwise, there will be no "consequences" of preparing or implementing the interventions. Also, because your title says "improving", which needs to be demonstrated.

We hope you are able to add these aspects and resubmit with the required clarifications, changes and additions.

Reviewers' comments:

Reviewer's Responses to Questions

**Comments to the Author**

1. Is the manuscript technically sound, and do the data support the conclusions?

Reviewer #1: Yes

Reviewer #2: Yes

2. Has the statistical analysis been performed appropriately and rigorously? 

Reviewer #1: N/A

Reviewer #2: Yes

3. Have the authors made all data underlying the findings in their manuscript fully available?

Reviewer #1: No

Reviewer #2: Yes

4. Is the manuscript presented in an intelligible fashion and written in standard English?

Reviewer #1: Yes

Reviewer #2: Yes

5. Review Comments to the Author

Reviewer #1: Authors described very well the methodology for evaluation the intervention, but is not completely clear the nature of the intervention, timing and implementation plans. It is clear that the evaluation was positive, but still need a better description of the intervention.

Reviewer #2: It is necesary to rewrite abbreviations in full statements in the introduction portions of the abstract ( such as SRH)On line number 34-36, please indicate how these participants (30 IDI and 18 FGD) were selected (random or purposeful).The introduction section was well-written and described. Further, the author’s need to incorporate the gab and consequence of this issue, specifically in the study settings.Is there any information about institutional determinants of SRH? Why the authors couldn’t want to mention barriers to accessing SRH services? Is only individual’s perceptions could affect SRH?What are the strengths of your study?Based on your conclusion, what is the major recommendation of responsible body?

6. PLOS authors have the option to publish the peer review history of their article (what does this mean?). If published, this will include your full peer review and any attached files.

Reviewer #1: No

Reviewer #2: No

---

## [Author Response · Author response to Decision Letter 0]

23 Nov 2023

Reviewer #1: 

Reviewer’s comment: Authors described very well the methodology for evaluation the intervention, but is not completely clear the nature of the intervention, timing and implementation plans. It is clear that the evaluation was positive, but still need a better description of the intervention.

Authors’ response: Details of the intervention plans and procedures had been earlier published by Mbachu et al., 2020 and referenced in this paper [22] (Page 8, line 178).

Reviewer #2: 

Reviewer’s comment: It is necesary to rewrite abbreviations in full statements in the introduction portions of the abstract (such as SRH)

Authors’ response: The abbreviation SRH in the introduction section of the abstract is written in full statement (Page 2, line 27)

Reviewer’s comment: On line number 34-36, please indicate how these participants (30 IDI and 18 FGD) were selected (random or purposeful).

Authors’ response: Thank you for the input. The study participants were purposively selected; this has been included (Page 2, line 38)

Reviewer’s comment: The introduction section was well-written and described. 

Further, the author’s need to incorporate the gab and consequence of this issue, specifically in the study settings.

Is there any information about institutional determinants of SRH? 

Why the authors couldn’t want to mention barriers to accessing SRH services? 

Is only individual’s perceptions could affect SRH?

Authors’ response: Thank you for the commendation.

The major institutional determinant of SRH in relation to the acceptability of the intervention was limited coverage, which was attributed to the paucity of funds. Other general determinants noted at need assessment were inadequacy in the number and capacity of health providers and poor community support to adolescents’ utilization of SRH services (Page 17, line 367-375. 

However, the barriers to accessing SRH services were not detailed because they are outside the scope of this paper. The paper focuses on the acceptability of the community-embedded intervention approach in improving SRH.

Reviewer’s comment: What are the strengths of your study?

Authors’ response: The strength of the study is that the collaboration between health and non-health sectors created multiple platforms for greater reach of adolescents with SRH services and facilitated knowledge learning and sharing, mutualistic relations, and networking across stakeholders and sectors, which would ensure the longevity of the approach and ownership of the program. This has been included (Page 33, lines 753-775)

Reviewer’s comment: Based on your conclusion, what is the major recommendation of responsible body?

Authors’ response: Considering the positive outcomes, it is recommended that the stakeholders consolidate the gains and acceptance of the collaborative intervention strategies and scale up the intervention. Furthermore, to mitigate the perceived fund constraints, collaboration with more partners is recommended for the continuity of the programs (Page 34, lines 771-774)

---

## [Editor Report · Decision Letter 1]

29 Nov 2023

Acceptability of a community-embedded intervention for improving adolescent sexual and reproductive health in south-east Nigeria: a qualitative study

PONE-D-23-09502R1

Dear Dr. Eze,

We’re pleased to inform you that your manuscript has been judged scientifically suitable for publication and will be formally accepted for publication once it meets all outstanding technical requirements.

Kind regards,

Alfredo Luis Fort, M.D., M.Sc., Ph.D.

Academic Editor

PLOS ONE

Additional Editor Comments (optional):

Good to see that the authors have looked into the comments made by reviewers and myself, and have added key clarifications/statements to complement or improve the significance of methods and results. Well done.